# The Environment Is a Key Factor in Determining the Anti-Amyloid Efficacy of EGCG

**DOI:** 10.3390/biom9120855

**Published:** 2019-12-11

**Authors:** Tomas Sneideris, Andrius Sakalauskas, Rebecca Sternke-Hoffmann, Alessia Peduzzo, Mantas Ziaunys, Alexander K. Buell, Vytautas Smirnovas

**Affiliations:** 1Institute of Biothechnology, Life Sciences Center, Vilnius University, LT-10257 Vilnius, Lithuania; 2Institute of Physical Biology, Heinrich-Heine-University, 40225 Düsseldorf, Germany; 3Department of Biotechnology and Biomedicine, Technical University of Denmark, DK-2800 Kgs Lyngby, Denmark

**Keywords:** insulin, α-synuclein, inhibition, EGCG, amyloid aggregation

## Abstract

Millions of people around the world suffer from amyloid-related disorders, including Alzheimer’s and Parkinson’s diseases. Despite significant and sustained efforts, there are still no disease-modifying drugs available for the majority of amyloid-related disorders, and the overall failure rate in clinical trials is very high, even for compounds that show promising anti-amyloid activity in vitro. In this study, we demonstrate that even small changes in the chemical environment can strongly modulate the inhibitory effects of anti-amyloid compounds. Using one of the best-established amyloid inhibitory compounds, epigallocatechin-3-gallate (EGCG), as an example, and two amyloid-forming proteins, insulin and Parkinson’s disease-related α-synuclein, we shed light on the previously unexplored sensitivity to solution conditions of the action of this compound on amyloid fibril formation. In the case of insulin, we show that the classification of EGCG as an amyloid inhibitor depends on the experimental conditions select, on the method used for the evaluation of the efficacy, and on whether or not EGCG is allowed to oxidise before the experiment. For α-synuclein, we show that a small change in pH value, from 7 to 6, transforms EGCG from an efficient inhibitor to completely ineffective, and we were able to explain this behaviour by the increased stability of EGCG against oxidation at pH 6.

## 1. Introduction

The onset and progression of more than 50 human disorders, including the neurodegenerative Alzheimer’s and Parkinson’s diseases (AD and PD), is associated with the failure of peptides and proteins to adopt or remain in their native functional and soluble states, and their subsequent conversion into amyloid fibrils [1,2]. Millions of people around the world suffer from these disorders; AD alone affects 40 million patients worldwide and is projected to rise steadily to afflict 135 million people by 2050 [3,4]. Distinct peptides and proteins are associated with these particular human disorders; however, the formation and accumulation of insoluble fibrillar aggregates are common among these diseases [1,2]. Whether extracted from patients or generated in vitro, amyloid fibrils formed from different proteins seem to be remarkably similar in overall morphology. Mature amyloid fibrils tend to appear as unbranched, thread-like, elongated structures, several nanometres in diameter and with lengths of the order of micrometres [1,2]. In addition, the corresponding fibrils all contain a β-sheet-rich structure, termed “cross-β,” according to the pattern in X-ray fibre diffraction studies [5].

Several therapeutic approaches, such as a reduction in the production of amyloidogenic peptides, the increase of the native state stability of amyloidogenic proteins, an enhancement in the clearance rates of misfolded proteins, and a direct inhibition of the self-assembly process, have been suggested for treatment of amyloid-related disorders [6,7,8,9,10,11,12]. Numerous small molecular weight compounds, short peptides, and antibodies have been suggested as potential modulators and inhibitors of toxic oligomeric and fibrillar species’ assemblies [6,8,9,12,13,14,15]. Despite significant and persistent efforts, there are still no effective disease-modifying drugs or treatment modalities available for the majority of amyloid-related disorders (overall success rate of clinical trials is <0.5%) [16,17,18,19,20,21].

The formation of amyloid fibrils is a complex process, which involves several microscopic steps (e.g., nucleation, growth, fragmentation, and secondary nucleation) [1,2]. Alterations in environmental conditions can modulate these microscopic steps, resulting in different pathways and leading to the formation of structurally distinct amyloid aggregates [1,22,23,24,25]. Such conformational variability, also known as polymorphism, is thought to be a generic property of amyloid proteins, and has been proposed to be able to cause distinct disease phenotypes [1,26,27]. Moreover, the effects of potential therapeutic agents can vary depending on the conditions in which they are assayed (i.e., pH, temperature, buffer, interfaces, agitation, and others) [12], possibly due to chemical modifications of potential therapeutic molecules [28,29,30]. Since the environmental conditions under which aggregation of amyloid proteins is performed can vary between distinct studies, the search and assessment of potential inhibitors becomes extremely challenging, as the results may lead to diverse conclusions.

An ideal amyloid assembly inhibitor should act as broadly as possible and be capable of interacting with different species along the pathway of aggregation. The desired outcome of such an interaction is to block the formation of toxic oligomeric species and to possibly dissociate preformed fibrillar aggregates into non-toxic species [31]. Epigallocatechin-3-gallate (EGCG), the main polyphenol found in green tea, has been reported to effectively inhibit the aggregation of a number of amyloidogenic peptides and proteins, including amyloid-β (related to AD) [32,33], α-synuclein (related to PD) [33,34,35,36], islet amyloid polypeptide (related to type-II diabetes) [37,38], huntingtin exon 1 (related to Huntington’s disease) [39], tau (related to AD and tauopathies) [40], superoxide dismutase (related to amyotrophic lateral sclerosis) [41], prion proteins (related to prion diseases) [42], and others. In addition, it has been shown that EGCG can induce remodeling and/or dissociation of pre-existing aggregate species [33,34,36,43,44]. Taken together, EGCG appears to be a “universal” inhibitor of amyloid fibril formation, suggesting that this molecule could be used as a therapeutic agent for the prevention and treatment of amyloid-related disorders. However, EGCG is not stable at neutral or alkaline pH [45,46,47,48], where it is susceptible to auto-oxidation, resulting in the formation of numerous products [28] which may differently affect amyloid aggregation [29,30,49].

In this study, we set out to explore whether or not the universal nature of EGCG as an inhibitor is robust against variation in solution conditions. We chose two amyloid forming protein systems, a model system (insulin) and a disease-related protein (α-synuclein), which form amyloid fibrils under very different solution conditions. This choice allowed us to separately probe the influence of EGCG oxidation and the interplay between the solution conditions and the action of EGCG.

The formation of insulin amyloid fibrils in vivo is associated with the clinical syndrome injection-localised amyloidosis, which was observed in diabetes patients after continuous subcutaneous injections of insulin [50,51]. Despite its main application in medicine, recombinant human insulin is also extensively used as a model protein to study pathways and mechanisms of amyloid fibril formation in vitro. It has been demonstrated that several sets of conditions, including the presence of ethanol [52,53,54]; different pHs [22,55] or salt concentrations [23,55]; and agitation [55,56], can alter insulin aggregation pathways and even lead to the formation of structurally different amyloid fibrils. The majority of insulin aggregation studies were performed at low pHs, which do not reflect the physiological environment, but serve to significantly accelerate amyloid fibril formation through a destabilisation of the native state. At the same time, acidic conditions are known to lead to higher EGCG stability against oxidation. We demonstrate the different effects of EGCG and its auto-oxidation products (EGCG${}_{ox}$) on insulin aggregation. Furthermore, we show that even under the acidic reaction conditions where EGCG is stable, the exact solvent conditions determine whether or not EGCG is able to modulate the kinetics of insulin amyloid fibril formation.

The strong pH-dependence of EGCG’s stability against oxidation in the proximity of neutral pH enabled us to probe the interplay of protein aggregation and EGCG oxidation, using the protein α-synuclein, associated with Parkinson’s disease [57]. By comparing α-synuclein amyloid fibril formation at pH 7, where EGCG rapidly oxidises, and pH 6, where it is much more stable, we found that EGCG converts from an efficient inhibitor at pH 7 to being completely ineffective at pH 6. On the other hand, pre-oxidised EGCG is a very powerful inhibitor at pH 6 as well. Taken together, in this study we demonstrate that even an inhibitor candidate as well-characterised as EGCG can display a dramatically different inhibitory efficiency depending on the solution conditions, and hence a systematic exploration of the interplay of solution conditions and compound stability and efficacy is crucial.

## 2. Results

We first performed insulin amyloid fibril formation experiments under different acidic solution conditions. When the insulin aggregation reaction was performed in 100 mM phosphate buffer, pH 2.4 (PB), under quiescent conditions, the presence of EGCG increased the half-time ($t_{50}$), i.e., the time to reach half the maximal Thioflavin-T (ThT) fluorescence intensity, by almost two-fold, while at the same time decreasing the maximum fluorescence intensity (*I${}_{max}$*) nearly two-fold, when compared to the control (Figure 1 and Appendix A). The effect of EGCG${}_{ox}$ (see Methods section for experimental protocol to generate oxidised EGCG) is stronger, and leads to an almost four times longer $t_{50}$ and almost four times lower *I${}_{max}$*. In the Appendix A, we show time-resolved UV–Vis data of EGCG that demonstrates the lack of oxidation under the conditions of these kinetic experiments (Appendix A). Under agitated conditions in PB, EGCG has no inhibitory effect, while EGCG${}_{ox}$ has a minor effect on the insulin aggregation process (Figure 1 and Appendix A).

The presence of EGCG${}_{ox}$ results in a two times longer $t_{50}$ and 20 times higher *I${}_{max}$* effect, when the aggregation reaction is performed in 20% acetic acid (AC), under quiescent conditions (Figure 1). When agitation is applied, the presence of EGCG${}_{ox}$ results in a three times higher *I${}_{max}$* and has a minor effect on $t_{50}$ (Figure 1 and Appendix A). The presence of non-oxidised EGCG has no effect on $t_{50}$ or *I${}_{max}$*, when the aggregation reaction is performed under either quiescent or agitated conditions (Figure 1 and Appendix A) in AC. Taken together, these results suggest that under these sets of solution conditions, EGCG has only a weak effect on insulin amyloid fibril formation, which is reinforced by oxidation of EGCG. Furthermore, the results also suggest that the absolute fluorescence intensity of ThT bound to insulin amyloid fibrils is strongly influenced by the presence of EGCG${}_{ox}$.

Sample analysis using atomic force microscopy (AFM) confirmed the formation of insulin amyloid fibrils within 15 h under all test conditions (Figure 2, Appendix A). Typically, individual fibrils are 3–10 nm in height and their lengths range from several hundred nm to several μm. In PB, insulin amyloid fibrils tend to cluster, and larger bundles were apparent when the reaction was performed under agitated conditions. In the presence of EGCG${}_{ox}$, the fibrils seem to be more dispersed (Figure 2, Appendix A). In AC more fibrils can be seen in the presence of EGCG${}_{ox}$, when compared to the control sample (Figure 2, Appendix A), even though care must be taken when quantitatively comparing AFM images and correlating these results with the composition of the solution. Fibrils formed in the presence of EGCG under all environmental conditions are similar in morphology to their respective control samples; i.e., the absence of EGCG or EGCG${}_{ox}$.

The secondary structure of insulin amyloid fibrils was assessed using Fourier-transform infrared (FTIR) spectroscopy (Figure 3). Second derivative FTIR spectra of fibrils formed in AC under quiescent and agitated conditions are almost identical, both showing a major minimum at 1627 cm${}^{-1}$ and a minor one at 1641 cm${}^{-1}$ in the amide I/I’ region, attributed to β-sheet structure and an additional band at 1729 cm${}^{-1}$ (Figure 3), which was assigned to the stretching vibrations of a deuterated carboxyl group (-COOD) [58]. Similarly, a major minimum at 1627 cm${}^{-1}$ in the amide I/I’ region, is present in case of PB under agitated conditions; however, the other two minima observed in AC are missing. The second derivative FTIR spectrum of insulin amyloid fibrils formed in PB under quiescent conditions has two minima at 1625 cm${}^{-1}$ and 1637 cm${}^{-1}$ in the Amide I/I’ region. It confirms that fibrils formed without agitation in PB are structurally different from fibrils formed in AC, while the fibrils formed in PB with agitation seem to have a secondary structure profile, which looks like an intermediate between PB and AC. These results suggest that despite the very similar morphology, as judged from AFM images, the insulin amyloid fibrils formed under different solvent conditions have some structural differences.

The insulin aggregation experiments under acidic conditions described above allow one to isolate the oxidation of EGCG from the protein aggregation. However, in many cases, amyloid fibril formation is studied under conditions under which EGCG is highly unstable. We therefore performed additional amyloid fibril formation experiments with α-synuclein, the aggregation of which is associated with Parkinson’s disease [57]. α-synuclein forms amyloid fibrils at both neutral and mildly acidic pH [25], which provides an ideal paradigm to study the inhibition by EGCG, because the latter compound displays a dramatic change in stability between neutral and slightly acidic pH (Appendix A). We incubated monomeric α-synuclein in polystyrene plates under shaking, and in the presence of glass beads. Under these conditions, the surface-catalysed nucleation [59] and subsequent amplification through fragmentation [25] of α-synuclein amyloid fibrils is very efficient. We compared the time course of ThT fluorescence at pH 7, where it has been shown that EGCG is an efficient inhibitor of α-synuclein amyloid fibril formation [33], and at the slightly more acidic pH of 6 (Figure 4A,B). We found that, based on ThT intensity alone, at pH 7, EGCG completely inhibits the formation of α-synuclein amyloid fibrils at a stoichiometric ratio of 1:1, whereas at pH 6, the maximal ThT intensity is merely reduced by a factor of two, while the half time is very similar compared to the absence of EGCG.

AFM images (Figure 5) show amyloid fibrils at pH 7 without EGCG and at pH 6 both in the presence and absence of EGCG. At pH 7 in the presence of EGCG, AFM imaging reveals some amorphous aggregates together with very short fibrillar structures, and in the presence of EGCG${}_{ox}$, almost no fibrils are found. The situation is dramatically different at pH 6, where fibrils can clearly be seen under all conditions, albeit very few in the presence of EGCG${}_{ox}$, where ThT fluorescence is completely suppressed. In order to obtain an independent measure for the degree of inhibition of aggregation by EGCG and EGCG${}_{ox}$, we centrifuged the samples after the aggregation experiment and quantified the average size and concentration (Figure 4C) of the soluble protein by microfluidic diffusional sizing (MDS) [60,61] (see Methods section for details). We found that in the absence of EGCG, both at pH 7 and pH 6, the protein converts near-quantitatively into aggregates, whereas in the presence of EGCG and EGCG${}_{ox}$, nearly all of the protein remains soluble, and display average hydrodynamic radii of ∼2.3 nm at pH 6 and ∼2.7 nm at pH 7, indistinguishable from measurements of pure monomeric protein and in close agreement with previous measurements under similar solution conditions [62]. Interestingly, at pH 6, MDS reveals that EGCG has no effect on the conversion efficiency into aggregates, and even in the presence of EGCG${}_{ox}$, inhibition is only partial, despite the fact that ThT fluorescence is completely quenched. We also accompanied these experiments by UV–Vis spectroscopic stability studies of EGCG under the same solution conditions, and we found that while the EGCG absorption spectrum undergoes substantial changes at pH 7 already after 1 h, almost no change is observed at pH 6 after almost 1 day of incubation (Appendix A).

## 3. Discussion and Conclusions

The effects of potential inhibitor compounds on the process of amyloid fibril formation are often determined by analysing the kinetics of aggregation [15,29,30,32,33,63,64,65,66,67,68] and/or the maximum ThT intensity [29,30,32,33,65,67,68,69,70,71] in the absence and presence of the compound. The effects of EGCG and EGCG${}_{ox}$ on the process of amyloid fibril formation by both insulin and α-synuclein performed under distinct environmental conditions were assessed using both aforementioned approaches (Figure 1 and Appendix A), and the conclusions are presented in (Table 1). In the case of insulin, if $t_{50}$ and/or $I_{max}$ were used as the main criteria, EGCG could be defined as an inhibitor of amyloid formation only if the screening was performed in PB under quiescent conditions. In case of EGCG${}_{ox}$ the picture is more complex. In PB, EGCG${}_{ox}$ was found to be an inhibitor independently of the assessment criteria, whereas in AC, $t_{50}$ points towards an inhibitory effect, while $I_{max}$ suggests an enhancement of aggregation. In the case of α-synuclein amyloid fibril formation, on the other hand, both criteria suggest EGCG to be a strong inhibitor at pH 7, whereas only $I_{max}$ indicates inhibition at pH 6. In the latter case, only the inclusion of the soluble protein at the end of the reaction as an additional measured parameter allows to correctly evaluate the inhibitory effect. These results suggest that depending on aggregation conditions and the screening criteria, the same compound could be defined as a hit or a failure. This raises the question as to the origin of such variable results.

First, alterations in environmental conditions can modulate protein aggregation pathways and result in the formation of structurally distinct amyloid aggregates (Figure 6A) [22,23,24,26,27]. Thus, it is plausible that species targeted by the compound might exist only under certain environmental conditions. Indeed, EGCG inhibits the insulin aggregation reaction only when the latter is performed in PB under quiescent conditions. AFM analysis did not reveal any major differences between fibrils formed in the absence or presence of EGCG (Figure 2). However, differences in the secondary structure of fibrils, determined using FTIR (Figure 3), suggest the possibility of distinct pathways and intermediates involved in the process of insulin fibril formation in PB under quiescent or agitated conditions or in AC under both the presence and absence of agitation. It is possible that the molecular species targeted by EGCG or its oxidation products are only present under certain environmental conditions. A similar explanation can be valid for the different relative $t_{50}$ values in PB and AC in the presence of EGCG${}_{ox}$. The strong increase in ThT fluorescence intensity in the presence of EGCG${}_{ox}$ in AC, on the other hand, requires an alternative explanation. A simple increase in the quantity of fibrils formed is not sufficient to explain the observed several-fold increase in $I_{max}$. It has been demonstrated before that amyloid fibrils formed under distinct environmental conditions may possess different ThT binding sites, affinities for ThT, and ThT quantum yields [72,73,74]. Thus, since the secondary structure of insulin amyloid fibrils formed in PB and AC was found to be different (Figure 3), it is possible that EGCG${}_{ox}$ induces slight conformational changes in the amyloid fibrils formed in AC, which results in an increase in quantum yield in the bound ThT, and therefore, in increased ThT fluorescence intensity. However, no obvious differences in morphology or secondary structure (Appendix A), of insulin fibrils formed in AC in the absence or presence of EGCG${}_{ox}$ were observed. Therefore, it is also possible that the change in ThT intensity stems from a direct interaction between bound ThT and EGCG${}_{ox}$. The fact that extrinsic compounds can dramatically change the ThT fluorescence quantum yield has sometimes led to false interpretation of a given compound as an inhibitor (Figure 6B). It has, for example, been shown that the two amyloid dyes, Congo red (CR) and ThT, have an affinity for each other and that CR strongly quenches ThT fluorescence, rather than inhibiting amyloid fibril growth [75]. Therefore, absolute fluorescence intensity is often not a reliable readout for the extent of inhibition by any given compound. This conclusion is further supported by the results obtained for α-synuclein at pH 6, where the final ThT intensity in the presence of EGCG suggests a significant inhibition, but measurement of soluble protein and AFM show that the sample has quantitatively been converted into fibrils. The nature of the surfaces involved (cuvette, plate surface, stir bar, and air-water-interface), in combination with the physico-chemical properties of the protein can also have a large impact on the protein aggregation process [59,76,77,78,79]. Indeed, additional experiments showed that the strong increase in $I_{max}$ observed in the presence of EGCG${}_{ox}$ depends on the type of surface of the microplate used (Appendix A). Under agitated conditions the effect of EGCG${}_{ox}$ on insulin aggregation was found to be weaker when compared to the one under quiescent conditions. Agitation in general speeds up amyloid fibril formation, mostly because of its effect on fibril fragmentation, and the detachment of species from the air-water or solid-water interface, where proteins have a strong tendency to accumulate and where in many cases the nucleation step of amyloid fibril formation is likely to occur. By selectively enhancing individual steps, such as fragmentation or nucleation, the concentrations of species that can be targeted by EGCG may be decreased, and hence, its inhibitory effect diminished.

Second, specific environmental conditions may induce modifications of compounds (Figure 6C) [28,29,30,80]. For example, EGCG is not stable at neutral pH and oxidises within several hours. In general, the effect of EGCG${}_{ox}$ on insulin aggregation, is stronger when compared to its non-oxidised form. A further striking example of the effect of the solution conditions on the inhibitory effects of EGCG is given by our findings that a change in pH by only one unit dramatically changes the inhibition of α-synuclein amyloid fibril formation. At the most often employed neutral pH of 7, where EGCG is highly unstable, almost complete inhibition is observed by stoichiometric amounts of EGCG, as evaluated by ThT fluorescence and microfluidic diffusional sizing (MDS). At the same time, UV–Vis experiments with EGCG under equivalent conditions show that EGCG undergoes rapid and quantitative oxidation within a similar time scale as the aggregation process itself (Appendix A and Figure 6C). This leads to the fact that mostly oxidised EGCG is available to inhibition. The effect of EGCG and EGCG${}_{ox}$ at pH 7 is to keep the protein in its monomeric form, as has recently also been reported [81]. The amorphous aggregates that have been observed to be formed by α-synuclein in the presence of EGCG at neutral pH could stem in part from monomeric protein that clusters into amorphous structures during sample preparation for AFM or electron microscopy. We note that even under these conditions of near complete inhibition, as evaluated by ThT fluorescence and MDS, some short fibrils can be seen in AFM images, stressing the importance of the use of multiple experimental methods in order to obtain a complete picture of the inhibitory action. A change to pH 6, however, leads to an increased stability of EGCG (as confirmed by UV–Vis spectroscopy, Appendix A), which is paralleled by a strongly decreased inhibitory effect on α-synuclein aggregation. Indeed, despite the fact that ThT fluorescence intensity is decreased by approximately 50% in the presence of stoichiometric amounts of EGCG and quantitatively suppressed in the presence of stoichiometric amounts of EGCG${}_{ox}$, MDS and AFM suggest no inhibition by EGCG and only partial inhibition by EGCG${}_{ox}$. These results not only suggest an influence of EGCG and EGCG${}_{ox}$ on ThT fluorescence intensity (Figure 6B), but most notably a dramatic pH dependence of the inhibitory effect of EGCG, most likely related to EGCG stability, as discussed above. This finding is highly relevant and interesting, as α-synuclein experiences environments with reduced pH during its life cycle, such as endosomes and lysosomes [82,83]. Furthermore, it has been shown that the aggregation of α-synuclein is strongly enhanced at mildly acidic pH values [25], as found in such microenvironments, whereas at the same time EGCG appears to lose its inhibitory effect. In conclusion, we demonstrate here that the environmental conditions and the methods used for assessments of the effects of inhibitory compounds play an important role in the reliable identification of anti-amyloid compounds. Under certain circumstances the study design may define whether a given compound is found to be a hit or a failure. Therefore, assessing the effects and the intrinsic stability of compounds under a range of environmental conditions in vitro is essential for the further development of the lead compounds resulting in increased success rates for in vivo studies and clinical trials.

## 4. Materials and Methods

### 4.1. Materials and Solutions

Initial solutions of insulin (Sigma Aldrich, St. Louis, MO, USA, number 91077C) were prepared by dissolving 2 mg of dry insulin powder in 0.5 mL of 100 mM sodium phosphate buffer, pH 2.4, supplemented with 100 mM NaCl (PB) or 20% acetic acid, supplemented with 100 mM NaCl (AC). Concentration of insulin (M.W.—5808 Da, $\varepsilon_{280}$—6335 M${}^{-1}$ cm${}^{-1}$) was determined by measuring UV-absorption at 280 nm using NanoDrop 2000 (Thermo Fisher Scientific, Wolsom, MA, USA). Subsequently, insulin solutions were diluted to a final concentration of 2 mg/mL (344 μM) using PB or AC and supplemented with 200 μM of Thioflavin-T (ThT; Sigma Aldrich, number T3516) from a 10 mM ThT stock solution (in MilliQ water). For the insulin inhibition experiments, fresh solutions of 344 μM of EGCG (Sigma Aldrich, number 989-51-5) were prepared by dissolving EGCG in 100 mM sodium phosphate buffer pH 2.4, supplemented with 100 mM NaCl or in 20% acetic acid, supplemented with 100 mM NaCl, just before the experiment. EGCG${}_{ox}$ was prepared by dissolving 10 mM of EGCG in 10 mM phosphate buffer solution, pH 7.4, and incubation for 8 h at 60 ${}^{\circ }$C in a thermomixer MHR 23 (Ditabis, Pforzheim, Germany). Subsequently, it was diluted to a final concentration of 344 μM using PB or AC.

The α-synuclein in the pT7-7 vector was expressed in *Escherichia coli* BL21 (DE3) and purified as previously described [25]. As a last step, α-synuclein was purified by size-exclusion chromatography on an ÄKTA pure chromatography system (GE Healthcare) using a Superdex 200 Increase 10/300 GL (GE Healthcare) and 20 mM citric acid, pH 7, as an elution buffer. α-synuclein concentration was determined by measuring UV-absorption at 275 nm (extinction coefficient of 5600 M${}^{-1}$ cm${}^{-1}$). For the α-synuclein inhibition experiments, 5 mM solutions of EGCG (Tocris, Abingdon, UK, number 4524) were prepared by dissolving EGCG in dH${}_{2}$O. The solutions were frozen and stored at –20 ${}^{\circ }$C, after monitoring no difference between fresh and thawed EGCG. EGCG${}_{ox}$ was prepared by dissolving 10 mM of EGCG in 20 mM citric acid, pH 7, and incubation for 6 h at 60 ${}^{\circ }$C in a thermomixer. Subsequently, it was diluted to a final concentration of 5 mM, frozen and stored at –20 ${}^{\circ }$C.

### 4.2. Measurements of Aggregation Kinetics

Insulin: For the inhibition experiments, 344 μM solutions of EGCG or EGCG${}_{ox}$ were mixed with 344 μM insulin solutions in a 1:1 ratio. Three replicates of each solution were then pipetted into a nonbinding surface plate (NBS; Corning, Corning, NY, USA, number 3881). The plate was sealed using sealing tape (Nunc, Roskilde, Denmark, number 232701). Kinetics of insulin aggregation was monitored at 60 ${}^{\circ }$C without (quiescent conditions) and with continuous shaking (960 rpm; agitated conditions) by measuring ThT fluorescence emission intensity (excitation—440 nm; emission—480 nm) through the bottom of the plate using a Synergy H4 Hybrid Multi-Mode (Biotek, Winooski, VT, USA) microplate reader for 15 h (readouts were taken every 5 min under quiescent conditions and every 2 min under agitated conditions). Three independent measurements were performed for each sample.

α-synuclein: To study the effect of EGCG on the α-synuclein fibril formation, solutions of 25 μM of α-synuclein were prepared with EGCG or EGCG${}_{ox}$ solutions in a 1:1 ratio, 20 μM ThT and 150 mM citric acid at the wanted pH-value (pH 6 or pH 7). Three replicates of each solution were then pipetted into a high binding surface plate Costar (Corning, number 3601) and glass beads were added into the wells. The plate was sealed using SealPlate film (Sigma-Aldrich, number Z369667). Kinetics of amyloid formation were monitored at 37 ${}^{\circ }$C under continuous shaking (300 rpm) by measuring ThT fluorescence intensity through the bottom of the plate using FLUOstar (BMG LABTECH, Ortenberg, Germany) microplate reader (readouts were taken every 5 min).

The highest ThT fluorescence emission value within each curve was assumed to be $I_{max}$. Half-times ($t_{50}$) of the aggregation process were obtained as described by Nielsen et al. [55]. Briefly, experimental data was fitted using the following sigmoidal equation:(1)$$Y=y_{i}+m_{i}t+\frac{y_{f}+m_{f}t}{1+e^{-(\frac{t-t_{50}}{\tau })}},$$
where *Y* is the ThT fluorescence emission intensity, *t* is the time, and $t_{50}$ is the time when 50% of maximum ThT fluorescence intensity is reached. The initial baseline is described by $y_{i}+m_{i}t$ and the final baseline is described by $y_{f}+m_{f}t$.

### 4.3. Evaluation of EGCG and EGCG${}_{ox}$ Effects on the Insulin Aggregation Process

The effects of EGCG and EGCG${}_{ox}$ on the insulin aggregation process were determined by comparing experimental values of $t_{50}$ or $I_{max}$ of control samples with the ones determined in the presence of EGCG or EGCG${}_{ox}$ using one-way the one-way analysis of variance (ANOVA). *p* < 0.01 was accepted as statistically significant. The analysis was performed using OriginPro software.

### 4.4. Atomic Force Microscopy (AFM)

Insulin: Directly after the kinetic measurements, the samples were collected, and 20 μL of each sample was deposited on freshly cleaved mica and incubated for 1 min. Subsequently, the samples were rinsed with 1 mL of MilliQ water and dried under gentle airflow. Three-dimensional AFM maps were acquired using a Dimension Icon (Bruker) atomic force microscope operating in tapping mode, equipped with a silicon cantilever Tap300AI-G (40 N m${}^{-1}$, Budget Sensors) with a typical tip radius of curvature of 8 nm. High-resolution (1024 × 1024 pixels) images were acquired. The scan rate was 0.5 Hz. AFM images were flattened using SPIP (Image Metrology, Hessholm, Denmark) or NanoScope Analysis (Bruker, Billerica, MA, USA) software.

α-synuclein: AFM images were acquired directly after the aggregation kinetic measurements. In total, 10 μL of each sample was deposited onto freshly cleaved mica. After drying, the samples were washed 5 times with 100 μL of dH${}_{2}$O and dried under gentle flow of nitrogen. Three-dimensional AFM maps were obtained using a NanoScope V (Bruker) atomic force microscope equipped with a silicon cantilever ScanAsyst-Air (Bruker) with a tip radius of 2–12 nm. High-resolution (1024 × 1024 pixels) images were acquired. The scan rate was 0.9 Hz. AFM images were flattened using SPIP (Image Metrology) software.

### 4.5. Fourier-Transform Infrared (FTIR) Spectroscopy

Insulin fibrils were separated from buffer solution by centrifugation at 10,000× *g* for 30 min and subsequently resuspended in 1 mL of D${}_{2}$O; the procedure was repeated three times. Finally, fibrils were resuspended in 0.3 mL of D${}_{2}$O and sonicated for 1 min using Sonopuls 3100 (Bandelin, Berlin, Germany) ultrasonic homogeniser equipped with MS73 tip (using 50% of the power; total energy applied to the sample ∼1.12 kJ). Samples were deposited between two CaF${}_{2}$ transmission windows separated by 0.05 mm teflon spacers. The FTIR spectra were recorded using a Vertex 80v (Bruker) IR spectrometer equipped with a mercury cadmium telluride detector, at room temperature under vacuum (∼2 mBar) conditions. A total of 256 interferograms of 2 cm${}^{-1}$ resolution were averaged for each spectrum. The spectrum of D${}_{2}$O was subtracted from the spectrum of each sample. All spectra were normalised to the same area of amide I/I’ band (1700–1595 cm${}^{-1}$). All data processing was performed using GRAMS software.

### 4.6. Microfluidic Diffusional Sizing and Concentration Measurements

To measure the concentration of the soluble α-synuclein, the samples were centrifuged for 60 min at 16,100× *g* at 25 ${}^{\circ }$C using a centrifuge 5415 R (Eppendorf) directly after the kinetic measurements. The supernatant was taken, and 6 μL was pipetted onto a disposable microfluidic chip and measured with the FluidityOne (Fluidic Analytics, Cambridge, UK). FluidityOne is a microfluidic diffusional sizing (MDS, [60]) device, which measures the rate of diffusion under steady state, laminar flow. The protein concentration is determined by fluorescence intensity, as the protein is mixed with ortho-phthalaldehyde (OPA) after the diffusion, a compound which reacts with primary amines, producing a fluorescent compound [61].

### 4.7. Time Course of EGCG Oxidation

The oxidation of EGCG was followed by UV–Vis spectroscopy. Solutions of EGCG were prepared as described above. In total, 10 mM EGCG solutions were dissolved in 10 mM sodium phosphate buffer pH 7.4, corresponding to the conditions under which a stock solution of EGCG${}_{ox}$ for insulin experiments was produced, and in 20 mM citric acid, pH 7, corresponding to the conditions under which a stock solution of EGCG${}_{ox}$ for the α-synuclein experiments was produced. The oxidation of EGCG was carried out by incubating the solutions at 60 ${}^{\circ }$C for 0–22 h. Subsequently, the solutions were diluted and the spectra were recorded in the wavelength range between 250 nm and 500 nm in a UV-transparent plate (Corning, Corning, NY, USA, number 3679) using a Spark (Tecan, Mannedorf, Switzerland) microplate reader. To monitor the stability of EGCG under the used aggregation experiments, 172 μM EGCG in 100 mM NaCl, 100 mM sodium phosphate buffer, pH 2.4, 172 μM EGCG in 20% acetic acid, and 100 mM NaCl were incubated at 60 ${}^{\circ }$C for 0–22h, corresponding to the conditions of the insulin aggregation experiments, and 125 μM EGCG in 150 mM citric acid at pH 6 and pH 7 were incubated at 37 ${}^{\circ }$C for 0–22 h, corresponding to the conditions of the α-synuclein aggregation experiments.

## Figures and Tables

**Figure 1 biomolecules-09-00855-f001:**
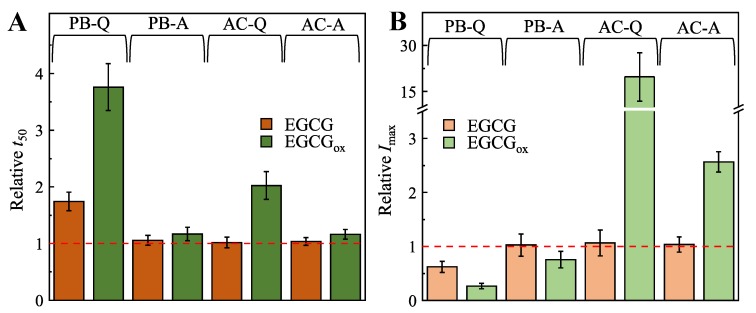
Effects of EGCG and EGCG${}_{ox}$ on insulin aggregation kinetics (**A**) and maximum ThT fluorescence intensity (**B**). Abbreviations PB and AC represent environmental conditions (100 mM phosphate buffer and 20% acetic acid, respectively), while Q and A denote the agitation conditions (quiescent and agitated, respectively), under which the insulin aggregation reactions were performed. Error bars represent standard deviations.

**Figure 2 biomolecules-09-00855-f002:**
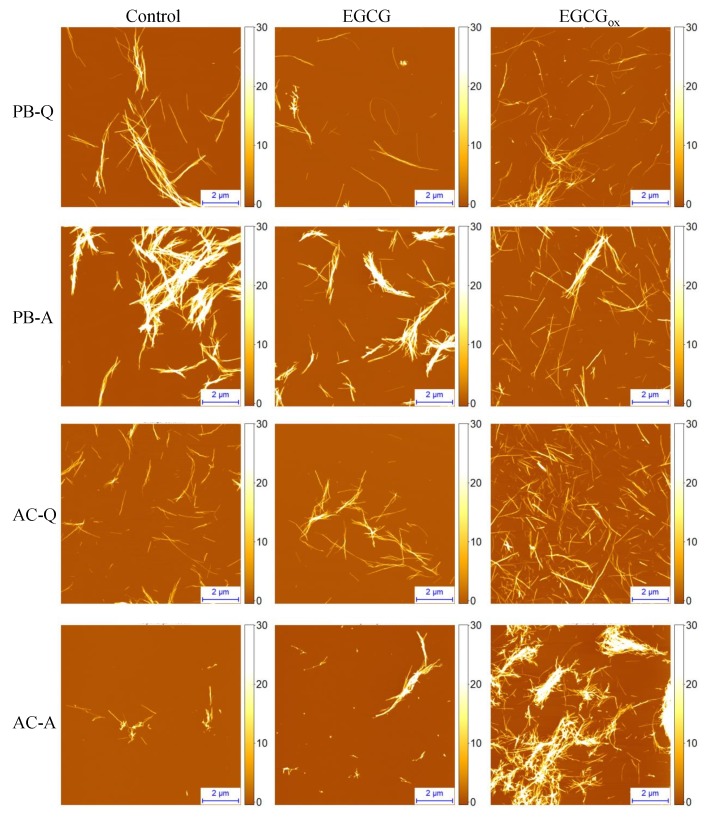
AFM images of insulin amyloid fibrils formed in PB or AC in the absence and presence of EGCG or EGCG${}_{ox}$. Abbreviations Q and A denote agitation conditions (quiescent and agitated, respectively), under which the insulin aggregation reactions were performed. The height scale (*z*) is in nm.

**Figure 3 biomolecules-09-00855-f003:**
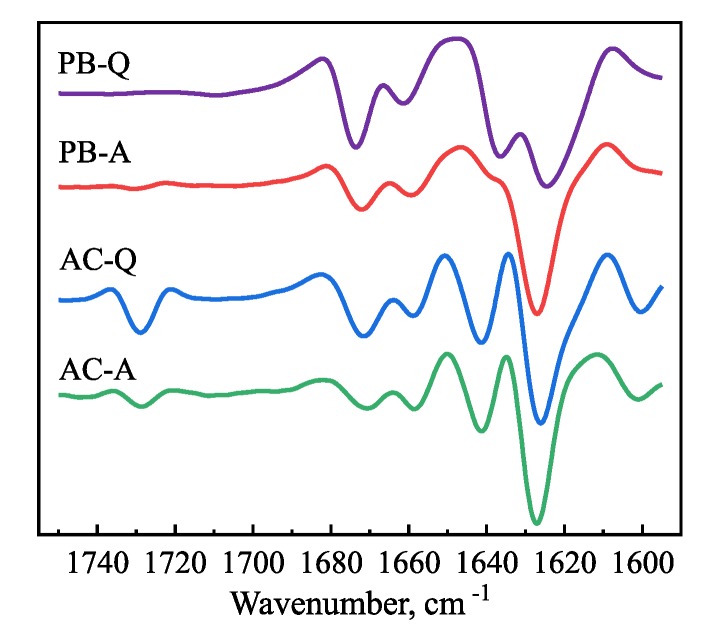
Second derivative FTIR spectra of insulin amyloid-like aggregates formed in PB and AC under quiescent and agitated conditions. Abbreviations PB and AC represent environmental conditions (100 mM phosphate buffer and 20% acetic acid, respectively), while Q and A denote agitation conditions (quiescent and agitated, respectively), under which the insulin aggregation reaction was performed.

**Figure 4 biomolecules-09-00855-f004:**
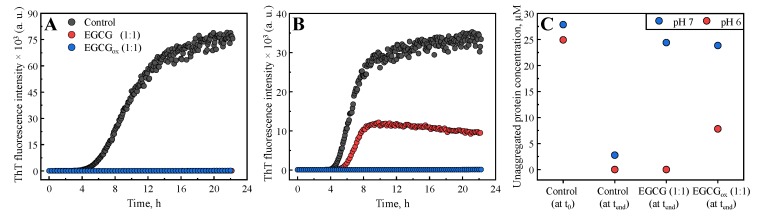
The effects of EGCG and EGCG${}_{ox}$ on the aggregation kinetics of α-synuclein monitored at pH 7 (**A**) and pH 6 (**B**). (**C**) α-synuclein concentration measured in the supernatant after centrifuging the end product of the aggregation reactions at pH 7 and pH 6, respectively.

**Figure 5 biomolecules-09-00855-f005:**
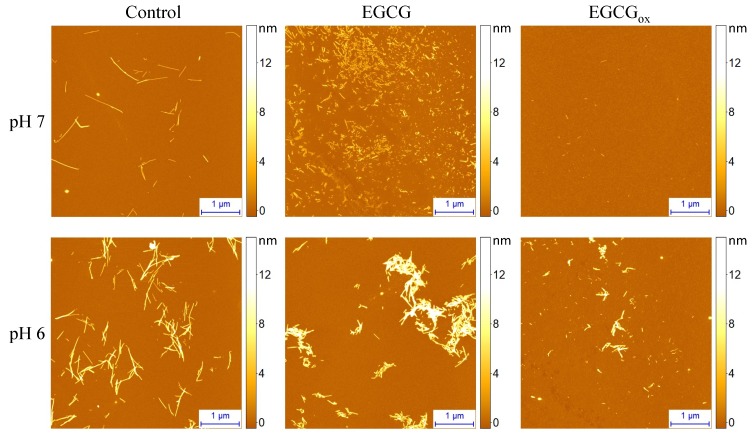
AFM images of α-synuclein aggregates formed at pH 7 or pH 6 in the absence and presence of EGCG or EGCG${}_{ox}$.

**Figure 6 biomolecules-09-00855-f006:**
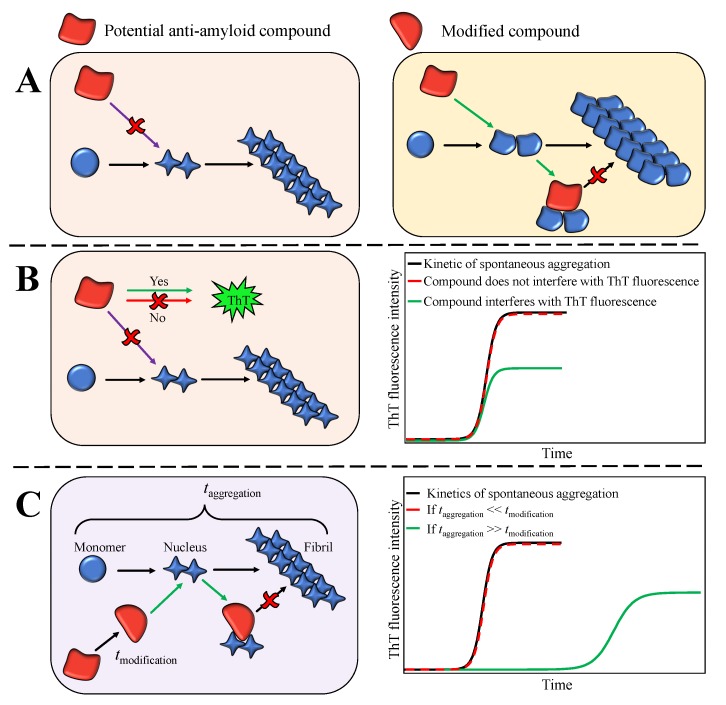
Schematic depiction of possible effects of potential anti-amyloid compounds on the amyloid aggregation reaction performed under distinct environmental conditions. Different environmental conditions can lead to the formation of distinct aggregate species of which only some are targeted by the compound (**A**), as in case of insulin aggregation in PB-Q and AC-Q. Certain compounds can also interfere with ThT’s fluorescence intensity (**B**), suggesting inhibition, which was not confirmed by other experiments, such as the quantification of soluble protein at the final plateau of ThT intensity. An example is EGCG and α-synuclein at pH 6. Moreover, specific environmental conditions can induce modifications of the compound, which results in the generation of products that target aggregation prone species (**C**), as in the case of α-synuclein aggregation at neutral pH. The compound modification can only manifest itself if it occurs with kinetics comparable to, or faster than the kinetics of aggregation. Distinct background colours represent different environmental conditions. Different shapes of aggregates represent distinct pathways reflecting the observed polymorphism of amyloid fibrils.

**Table 1 biomolecules-09-00855-t001:** Evaluation of EGCG and EGCG${}_{ox}$’s effects on the insulin aggregation process.

**Assessed by Change in** $t_{50}$			
**Protein**	**Conditions**	**EGCG**	**EGCG** ${}_{\mathrm{ox}}$
Insulin	PB-Q	Inhibitory ^1^	Inhibitory
	PB-A	No Effect	Inhibitory
	AC-Q	No Effect	Inhibitory
	AC-A	No Effect	Inhibitory
α-synuclein	pH 7	Inhibitory	Inhibitory
	pH 6	No Effect	Inhibitory
**Assessed by Change in** $I_{\max}$			
**Protein**	**Conditions**	**EGCG**	**EGCG** ${}_{\mathrm{ox}}$
Insulin	PB-Q	Inhibitory	Inhibitory
	PB-A	No Effect	Inhibitory
	AC-Q	No Effect	Enhancing
	AC-A	No Effect	Enhancing
α-synuclein	pH 7	Inhibitory	Inhibitory
	pH 6	Inhibitory	Inhibitory

${}^{1}$ Established by comparing experimental values of $t_{50}$ or $I_{max}$ of control samples with the ones determined in the presence of EGCG or EGCG${}_{ox}$ using one-way ANOVA (See Appendix A). *p* < 0.01 was accepted as statistically significant.

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
