# Peer review of "The Environment Is a Key Factor in Determining the Anti-Amyloid Efficacy of EGCG"

_biomolecules, 2019, doi:10.3390/biom9120855_

Round 1

Reviewer 1 Report

As authors mention in the introduction, the effect of EGCG on amyloid formation is well studied and reported, including oxidised and reduced forms as well as remodelling of pre-existing amyloid fibrils. Therefore, my main concern is the novelty of the study. 

Please also comment the relevance of low pH conditions chosen for insulin aggregation to physiological environment. Please comment on the choice of the phosphate buffer for pH=2.4, whereas PB performs the best at pH=5.8-8.0.

Authors specify that EGCG is not stable at neutral pH and auto-oxidises in a few hours. This implies that this molecule is mostly in oxidised form in the environment. Moreover, this also implies that the oxidative state of this chemical is hard to control. Taking these circumstances, please comment on the choice of using this anti-amyloidogenic agent as a model in this study.

Authors could get some quantitative information out of the FTIR spectra of the insulin amyloids. A spectrum of non-aggregated insulin could be included to the graph as an indicator of secondary structure transition.

Data presented in Fig. 1A is basically another representation of the results in panels C and D and is not necessary (or can be included in the SI).

Please explain different methods chosen for oxidised EGCG.

Please include more information about the AFM cantilever used for a-synuclein imaging and explain the range 2-12 nm.

Reviewer 2 Report

The paper entitled "Environment is a key factor in determining the efficacy of anti-amyloid compounds" deals with a very important issue related to environment conditions that play crucial roles in amyloid aggregation and in the identification of inhibitors of this process.

Even if the initial goal deserves attention, the development of the study is often unclear and confused and lacks of a clear message. The experiments appear well-conducted but they appear not homogeneous to each other.

The manuscript need a critical revision oriented toward the expressed message that in the present form is completely unclear.

Other issues need to be addressed as follows:

1)        Lines 96-97 which are the conclusions related to this sentence? Does the tht intensities related to oligomeric states? How?

2)        Lines 106-107 Better explain how are AFM images during time? How are they in comparison with other studies?

3)        There are no correlation and discussion in both model proteins on the choice of pH values, of stirring conditions. How are the protonation states of the proteins in different conditions? How can the authors compare the observed kinetic differences with the pIs of proteins? Which is the protonation state of EGCG and EGCGox?

4)        Lines 121-122 unclear. Rebuilt the sentence related to the choice of synuclein

5)        Line 170 Which reaction conditions? Are times of aggregation of both proteins comparable to those of oxidation of EGCG? Explain better it is completely unclear

6)        Figure 6 tries to summary the study but it is completely unclear for me: please define Env A and B (in A the compound is not inhibitor and in B is?? It is completely unclear!) Color and form codes need to be explained as well as the fields of each panel

7) The title must be changed since it refers to generic compounds but the study is carried out only on EGCG

Minor

Abstract needs to be clarified in the effective results, actually it appears too generic

Indicate Quisecent conditions also in the main text

Reviewer 3 Report

Manuscript ID: biomolecules-630740

Title: “Environment is a key factor in determining the efficacy of anti-amyloid compounds” by Sneideris et al.

Reviewer’s comments:

This study is aimed at understanding the effect of environmental (external) factors (e.g., pH, type of solvent/type of buffer) on the potency of anti-amyloidogenic compound, EGCG and oxidized EGCG, toward amyloid fibril formation of proteins, such insulin and alpha-synuclein, using several biophysical techniques and analytical tools including ThT binding assay for monitoring aggregation kinetics, FTIR for assessing secondary structural changes, and AFM for examining morphological features of the samples. Finally, authors concluded that the environmental conditions as well as the methods used for evaluation of the effects of inhibitory compounds are key to reliably identifying anti-amyloid compounds. As a result, assessing the effects of compounds under a range of environmental conditions in vitro is essential for the further development of the lead compound leading to increased success rates in in vivo studies and clinical trials. In addition, a schematic illustration delineating the influence of anti-amyloid compound on the formation of amyloid aggregates under different incubation conditions/environmental conditions was put forth.

The manuscript appears to be presented in a straightforward manner. This reviewer has some questions/suggestions regarding the current version of the manuscript.

This reviewer suggests that, apart from the general concluding remarks, authors should add some specific results/numbers in the abstract. There are some typographical errors and problems of word usage in the current version of the manuscript. To meet the quality of the journal, editorial editing/proofreading on the content of the manuscript is highly recommended. For completeness purposes. statistical analysis methods should be conducted and relevant procedure should be included in the Materials and Methods section. I wonder why the effects of potential anti-amyloid compounds (EGCG, EGCGox) on changes in tertiary structure were not examined in this study. For the sake of completeness, assays/experiments associated with cell cultures should be performed to gain some more insights into the influence of two anti-amyloid compounds on rescuing cells from amyloid fibril-induced cytotoxicity. This reviewer suggests that authors should include the comparison between the results and/or mechanisms obtained in this work and those reported in other relevant studies in the discussion section. 

Round 2

Reviewer 1 Report

The authors adequately addressed my questions and incorporated the answer into the manuscript. I recommend this manuscript for the publication in Biomolecules in its present form.